# Gp35/50 mucin molecules of *Trypanosoma cruzi* metacyclic forms that mediate host cell invasion interact with annexin A2

**Thiago Souza Onofre, Leonardo Loch, João Paulo Ferreira Rodrigues, Silene Macedo, Nobuko Yoshida** [ID] *

Departamento de Microbiologia, Imunologia e Parasitologia, Escola Paulista de Medicina, Universidade Federal de São Paulo, São Paulo, São Paulo, Brazil

* nyoshida@unifesp.br

**Data Availability Statement:** All relevant data are within the manuscript and its Supporting information files.

## Abstract

Host cell invasion is a critical step for infection by *Trypanosoma cruzi*, the agent of Chagas disease. In natural infection, *T. cruzi* metacyclic trypomastigote (MT) forms establish the first interaction with host cells. The gp35/50 mucin molecules expressed in MT have been implicated in cell invasion process, but the mechanisms involved are not well understood. We performed a series of experiments to elucidate the mode of gp35/50-mediated MT internalization. Comparing two parasite strains from genetically divergent groups, G strain (TcI) and CL strain (TcVI), expressing variant forms of mucins, we demonstrated that G strain mucins participate in MT invasion. Only G strain-derived mucins bound to HeLa cells in a receptor-dependent manner and significantly inhibited G strain MT invasion. CL strain MT internalization was not affected by mucins from either strain. HeLa cell invasion by G strain MT was associated with actin recruitment and did not rely on lysosome mobilization. To examine the involvement of annexin A2, which plays a role in actin dynamic, annexin A2-depleted HeLa cells were generated. Annexin A2-deficient cell lines were significantly more resistant than wild type controls to G strain MT invasion. In a co-immunoprecipitation assay, to check whether annexin A2 might be the receptor for mucins, protein A/G magnetic beads crosslinked with monoclonal antibody to G strain mucins were incubated with detergent extracts of MT and HeLa cells. Binding of gp35/50 mucins to annexin A2 was detected. Both G strain MT and purified mucins induced focal adhesion kinase activation in HeLa cells. By confocal immunofluorescence microscopy, colocalization of invading G strain MT with clathrin was visualized. Inhibition of clathrin-coated vesicle formation reduced parasite internalization. Taken together, our data indicate that gp35/50-mediated MT invasion is accomplished through interaction with host cell annexin A2 and clathrin-dependent endocytosis.

## Author summary

Host cell invasion by *Trypanosoma cruzi*, the agent of Chagas disease, is critical for the establishment of infection. Metacyclic trypomastigote (MT) forms are responsible for the

**Funding:** This work was supported by São Paulo Research Foundation (FAPESP) Grant 2016/15000-4, and in part by the Coordenação de Aperfeiçoamento de Pessoal de Nível Superior – Brazil (CAPES) - Finance Code 001. The funders played any role in the study design, data collection and analysis, decision to publish, or preparation of the manuscript.

**Competing interests:** The authors have declared that no competing interests exist.

initial *T. cruzi*-host cell interaction. Mucin molecules expressed on MT surface have been implicated in target cell invasion process, but the underlying mechanism are not fully understood. In this study, we aimed at elucidating the mode of mucin-mediated MT internalization. We found that requirement of mucins for MT invasion is *T. cruzi* strain-dependent. Experiments with G strain MTs, which rely on mucins and on target cell actin for internalization, revealed that mucin molecules bind to annexin A2, a protein that plays a role in actin dynamic. Annexin A2-deficient cell lines were generated and found to be significantly more resistant than wild type controls to MT invasion. Both MT and purified mucins induced focal adhesion kinase activation in host cells. By confocal immunofluorescence microscopy, invading MT was found to colocalize with clathrin, a protein that plays a role in endocytosis. Inhibition of clathrin-coated vesicle formation reduced parasite internalization. From these data we infer that mucin-mediated MT invasion is accomplished through interaction with host cell annexin A2 and clathrin-dependent endocytosis.

## Introduction

Annexins are a multigene family of $Ca^{2+}$-regulated phospholipid-binding proteins with diverse functions, with members of the family being expressed throughout animal and plant kingdoms [1,2]. Members of the annexin family have the requisite properties to integrate $Ca^{2+}$-signaling with actin dynamics at membrane contact sites [3]. Annexin A2 is an F-actin-binding protein enriched at actin assembly sites on the plasma membrane, and plays a role as a platform for actin remodeling [4–6].

*Trypanosoma cruzi*, the protozoan parasite that causes Chagas disease, invades the host cell in a manner dependent on $Ca^{2+}$-signaling [7,8] and actin cytoskeleton rearrangement [9–11]. Disruption of the target cell F-actin had no major effect on the internalization of tissue culture-derived trypomastigote (TCT) but inhibited the extracellular amastigote (EA) entry into cells [12,13]. EA internalization, which involves the formation of actin-rich membrane expansions resembling cup-like structures [14], was found to be reduced in annexin A2 knockout cells [15]. More recently, it was reported that TCT and EA enter target cells in a manner dependent on clathrin-mediated endocytosis [16], a process connected with actin cytoskeleton that participates in membrane dynamics [17,18].

Studies with metacyclic trypomastigote (MT) forms of *T. cruzi*, which are implicated in orally transmitted infection, responsible for outbreaks of acute cases of Chagas' disease in several countries in Latin America [19,20], have shown that different parasite strains induce distinct F-actin rearrangement during cell invasion [10,21]. For instance, the disassembly of host cell actin cytoskeleton did not affect invasion by CL strain but inhibited G strain internalization [10]. G and CL strains, which are classified as discrete typing units TcI and TcVI respectively [22], differ in their ability to invade target cells [23]. The highly invasive CL strain enters the host cell in a manner mediated by the MT stage-specific surface molecule gp82, which binds to lysosome-associated membrane protein LAMP2 [24,25], inducing lysosome spreading and exocytosis that contribute for the parasitophorous vacuole formation [26]. As regards G strain, it has been suggested that the mucin-like glycoproteins, gp35/50, may be implicated in MT entry into the host cell, based on the finding that the parasite internalization was inhibited by specific monoclonal antibody [27], but the mechanisms involved remain to be clarified. Gp82 and gp35/50 are structurally quite distinct glycoproteins. Exclusively expressed in MT [28], gp82 is encoded by a multigene family [29]. It contains N-linked oligosaccharides [30],

but the carbohydrate portion of the molecule is irrelevant for host cell adhesion or invasion [31]. The identity of amino acid sequences of gp82, as deduced from cDNA clones of G and CL strain MT, is 97.9% and, as regards the cell binding domain, the identity is 100% [23]. Gp35/50 molecules are expressed both in MT and epimastigotes [27]. They are encoded by a multigene family and are highly glycosylated proteins rich in threonine, with a unique type of glycosylation consisting of several sialylated *O*-glycans linked to the protein backbone via *N*-acetylglucosamine residues [32,33]. Abundantly expressed on the parasite surface, gp35/50 mucins are highly resistant to proteolytic degradation [34], a property associated with MT survival in the host stomach upon oral infection. G strain gp35/50 mucins bind to target cells in a manner mediated by receptor [8], which still awaits identification. Here we aimed at elucidating the mechanisms of host cell invasion by *T. cruzi* MT mediated by gp35/50 mucin molecules. Experiments were performed to unequivocally demonstrate the participation of gp35/50 mucins in G strain MT internalization and to determine whether they interacted with annexin A2. Human epithelial cells depleted in annexin A2 were generated and tested for the susceptibility to MT invasion. In addition, we examined the possibility that clathrin-dependent endocytosis and protein tyrosine kinase-mediated signaling were implicated.

## Methods

### Ethics statement

All procedures conformed to Brazilian National Committee on Ethics Research (CONEP) guidelines, and the study was approved by the Committee on Ethics of Animal Experimentation of Universidade Federal de São Paulo (protocol number: CEUA 9780200918). Biosecurity certificate (CQB) 028/97.

### Parasites, mammalian cells and invasion assay

*T. cruzi* strains G and CL were maintained alternately in mice and in liver infusion tryptose (LIT) medium containing 5% fetal bovine serum (FBS). To obtain cultures enriched in metacyclic forms, G strain was maintained in LIT medium up to the stationary growth phase, and CL strain was cultured for one passage in Grace's medium (Life Technologies/Thermo Fisher Scientific). For MT purification, the parasites were passed in DEAE-cellulose column, as described [28]. G strain EA forms were generated as follows: purified metacyclic forms were incubated with Vero cells for 24 h. After removal of unbound parasites, RPMI containing 1% FBS was added and incubation proceeded for 10–15 days, with change of medium every two to three days. Released trypomastigotes were collected, centrifuged and incubated for 12–14 h in LIT medium, pH 5.8, at 37°C, for differentiation into EA. Invasion assays were performed with human epithelial HeLa cells, as previously described [35], in RPMI medium containing FBS or in PBS$^{++}$ (PBS containing per liter: 140 mg CaCl$_2$, 400 mg KCl, 100 mg MgCl$_2$.6H$_2$O, 100 mg MgSO$_4$.7H$_2$O, 350 mg NaHCO$_3$), depending on the experiment. After 1 h incubation of HeLa cells with MT at MOI = 10 (CL strain) or MOI = 20 (G strain), or with EA at MOI = 5, the cells were fixed with Bouin's solution for 5 min, washed and stained with Giemsa solution (Sigma-Aldrich/Merck), followed by sequential dehydration with acetone, acetone:xylol, xylol. Giemsa-stained HeLa cell-coated coverslips were mounted on glass slides with Entellan (Merck Millipore), and the number of internalized parasites was quantified, by counting a total of 250 cells.

### Antibodies and reagents

Anti-LAMP2 antibody (H4B4) was from Developmental Studies Hybridoma Bank developed under the auspices of the NICHD and maintained by The University of Iowa, Department of

Biology, Iowa City, IA 52242. Antibodies to Annexin A2 (D11G2), β-tubulin (9F3), phospho-PKC (pan) (γThr514), phospho-p44/42 MAPK (ERK1/2) (Thr202/Tyr204) and phospho-tyrosine (p-Tyr-1000) were from Cell Signaling Technology. Antibody to phospho-FAK (Tyr397) was from Invitrogen and anti-clathrin antibody was from Sigma/Merck. TRITC-rhodamine phalloidin and Alexa Fluor 488-conjugated anti-mouse IgG were from Thermo Fisher Scientific, Alexa Fluor 555-conjugated anti-goat IgG was from Abcam. Monoclonal antibodies (mAbs) 5E7, 3F6, 10D8 and 2B10, directed to *T. cruzi* surface antigens, were produced and characterized in previous studies [27,28,34].

## Purification of *T. cruzi* mucin-like molecules

We followed the protocol used previously to purify mucins from epimastigotes and MT, which have essentially the same glycosylphosphatidyl anchor and the O-linked sugar chains [36]. A total of $5 \times 10^{10}$ parasites, from G or CL strain cultures containing mostly epimastigotes, which do not express gp82 [28], was centrifuged, the pellet was freeze-dried and placed in a sonicating water bath for 10 min with 10 ml of chloroform/methanol/water (1:2:0.8, by volume). After centrifugation at $2000 \times g$ for 5 min, and two more extraction of the pellet, the insoluble material served as source of delipidated parasites and the pooled fractions (30 ml) were placed in a round-bottom flask and dried by rotatory evaporation. The residue was extracted with 20 ml of butan-1-ol/water (2:1, by volume), the butan-1-ol phase contained the lipid fraction (F1) was discarded, the aqueous phase (F2) containing mucins was washed twice with water-saturated butan-1-ol and concentrated. The delipidated parasites were extracted three times by sonication with 10 ml of 9% butanol in water, and the pooled soluble material containing mucins (F3) was concentrated. The mucins containing F2 and F3 were resuspended in 2 ml of buffer A (0.1 M ammonium acetate in 5% propan-1-ol (v/v)) and fractionated on an octyl-Sepharose column, pre-equilibrated in buffer A. After washing the column with buffer A, and elution with a linear gradient at a flow rate of 12 ml/h, starting with 15 ml of buffer A and ending with 60% (v/v) propan-1-ol in water, fractions (2 ml) were analyzed by silver staining or Schiff staining of SDS-PAGE gels, as well as by immunoblotting using the monoclonal antibodies 2B10 and 10D8, directed to carbohydrate epitopes [34].

## Binding of purified gp35/50 mucins to HeLa cells

For cell binding assay, HeLa cells were seeded onto 96-well microtiter plates at $4 \times 10^4$ cells/well and were grown overnight at 37˚C. After fixation with 4% paraformaldehyde in PBS, washings with PBS and blocking with PBS containing 2 mg/ml BSA (PBS-BSA) for 1 h at room temperature, the cells were incubated for 1 h at 37˚C with purified mucin in PBS-BSA. Following washes and 1 h incubation with anti-mucin mAb in PBS-BSA, the cells were incubated with anti-mouse IgG conjugated to peroxidase. The bound enzyme was revealed using *o*-phenylenediamine and the absorbance at 490 nm was read in ELx800 microplate reader (BioTek).

## Generation of annexin A2-knockdown HeLa cell lines by lentiviral transduction

We followed the previous described protocol [24], using plasmids containing shRNA sequences targeted to Annexin A2 (Sigma Aldrich/Merck, clone IDs: TRCN0000296322, sequence TGAGGGTGACGTTAGCATTAC, and TRCN0000289781, sequence CGGGATG CTTTGAACATTGAA). The first sequence is predicted to bind to CDS region, and the second to 3'UTR region, leading to 98–99% gene silencing in human cells. Briefly, the lentiviral particles produced in HEK293T cells were filtered in 0.45 µm syringe filter, to remove cell debris,

and added to 6-well plates coated with HeLa cells. After 24 h incubation in the presence of 4 μg/ml polybrene, the cells were washed, incubated in R10 for 24 h, and then the medium was replaced every 48 h or 72 h, with increasing concentrations of puromycin, up to 10 μg/ml. The selected transduced HeLa cells were checked for Annexin A2 depletion by western blotting.

## Treatment of HeLa cells

Depletion of intracellular potassium was performed based on protocol described by Arkin et al. [37]. Briefly, HeLa cells were washed in buffer A (50 mM Hepes, 100 mM NaCl), pH 7.4, followed by incubation in hypotonic medium (RPMI/$H_2O$, 1:1) for 5 min at 37˚C, and in isotonic buffer A for 20 min. For cytoplasmic acidification, the procedure by Cosson et al. [38] was employed. HeLa cells were washed in RPMI/FBS (RPMI containing 20 mM Hepes, 4.5 g/ liter glucose, 10% FBS), and then incubated for 30 min at 37˚C in RPMI/FBS containing 20 mM $NH_4Cl$. Treatment with 0.45 M sucrose, or 10 μM chlorpromazine hydrochloride, consisted in incubating HeLa cells for 30 min with the drug in serum-free medium. HeLa cells were also treated for 45 min with FAK inhibitor PF573228 in serum-free medium.

## Indirect immunofluorescense assay

For confocal microscopy visualization of HeLa cell F-actin, lysosomes, clathrin and nucleus, upon interaction either with MT or with purified gp35/50 mucins, coverslips with adherent cells were processed as previously described [35] using TRITC-rhodamine phalloidin, anti-LAMP antibody, anti-clathrin antibody, Alexa Fluor conjugated IgG, and DAPI. For detection of MT, anti-mucin mAbs 10D8 and 2B10 were used. The coverslips were mounted in ProLong Gold (Invitrogen), and confocal images were acquired in Leica TCS SP8 laser-scanning microscope (Leica, Germany), at Instituto de Farmacologia e Biologia Molecular (INFAR), Universidade Federal de São Paulo, using oil immersion 63X objective. The images were processed and analyzed using Leica LAS AF (Leica, Germany) and Imaris (Bitplane) software. An average of 10 fields were checked for consistency with the representative images shown in Results.

## Preparation of HeLa cell extract and western blotting

HeLa cells were washed three times with PBS and submitted to mechanical lysis in 10 mM Tris pH 7.5 buffer, containing 1 mM EDTA, 100 mM NaCl, 1% nonionic detergent Igepal CA630, 10% glycerol, protease cocktail inhibitor, 2 mM $Na_3VO_4$ and 1 mM NaF. After 30 min on ice, followed by centrifugation, detergent soluble proteins were quantified, subjected to 10% SDS-PAGE and transferred to nitrocellulose membrane. For immunoblot analysis, the membranes were blocked with 5% skimmed milk powder in TBS-T (50 mM Tris-HCl, pH 7.5, 150 mM NaCl and 0.1% Tween 20). Following incubation with the primary antibody and HRP-conjugated secondary antibody diluted in blocking solution, and washings in TBS-T, the protein bands were revealed with Immobilon Western Chemiluminescent HRP Substrate (Merck) in Hyperfilm ECL (GE Healthcare).

## Co-immunoprecipitation assay

Protein A/G magnetic beads (Pierce Crosslink Magnetic IP/Co-IP Kit, Thermo Fisher Scientific), crosslinked to mAb 10D8 directed to MUC-G, were incubated for 1 h at room temperature, under agitation, with detergent-solubilized G strain MT extract, prepared by treating 3 x $10^8$ cells with PBS plus 0.5% Igepal, followed by centrifugation at 16,000 x g for 10 min. After washes and incubations with HeLa cell extract, bound proteins were eluted and submitted to western blotting analysis.

### Statistical analysis

The Student's *t* test (GraphPad Prism software Version 6.01) was employed to evaluate significance between cells subjected to treatment and their controls.

## Results

### *T. cruzi* G strain MT internalization is mediated by gp35/50 mucin molecules, depends on host cell actin cytoskeleton and is independent of lysosome mobilization

Previous studies suggested that gp35/50 mucin molecules are implicated in G strain MT invasion [10,27]. To clearly demonstrate the role played by gp35/50 mucins in G strain MT invasion, a series of experiments was performed, using the serum-free PBS$^{++}$ medium, which is suitable to comparatively analyze the infectivity of different *T. cruzi* strains and the factors involved in the invasion process [39–41]. In serum-containing full nutrient medium the invasion rate of *T. cruzi* strains, such as G, is very low because high amounts of surface molecules gp82 and gp90, which bind to the target cell, are also shed into medium and interfere with MT-host cell interaction, whereas in PBS$^{++}$ shedding is reduced [41]. Here we checked whether gp35/50 mucins were differentially released by G strain MT in RPMI medium containing 1% serum (R1) and in PBS$^{++}$. The western blot analysis of the conditioned medium from parasites incubated for 30 min in R1 or PBS$^{++}$ revealed lower levels of gp35/50 in PBS$^{++}$ (S1A Fig). In MT invasion assay, performed by incubating HeLa cells with parasites for 1 h in R1 or in PBS$^{++}$, the number of internalized parasites was about three-fold higher in PBS$^{++}$ (S1B Fig), a difference similar to that observed when the MT infectivity in medium containing 10% serum (R10) and PBS$^{++}$ was compared [39,41].

We examined the effect of gp35/50 mucin molecules purified from G strain (MUC-G) or CL strain (MUC-CL) on MT internalization. MUC-G and MUC-CL display some difference in the mobility in SDS-PAGE gel, and differ in their reactivity toward monoclonal antibodies 2B10 and 10D8 (Fig 1). Both MUC-G and MUC-CL are recognized by mAb 2B10, but reaction with mAb 10D8, presumed to recognize epitopes containing galactofuranose, is restricted to MUC-G [23]. HeLa cells were incubated for 1 h with G or CL strain MT in absence or in the presence of MUC-G or MUC-CL, at 40 μg/ml, and the number of intracellular parasites was quantified. MUC-G, but not MUC-CL, significantly inhibited G strain MT invasion, whereas CL strain MT entry into HeLa cell was not affected by mucins, either from G or CL strain (Fig 2A). To

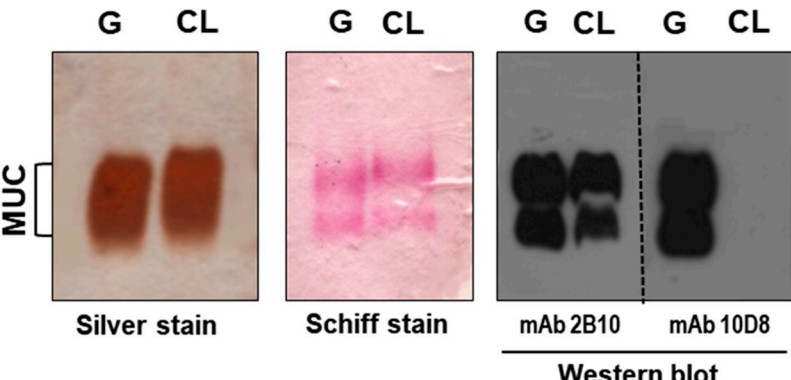

**Fig 1. Profile of purified gp35/50 mucins from *T. cruzi* strains G and CL.** Purified mucins were analyzed by silver staining or Schiff staining of SDS-PAGE gel, or by western blot using the indicated gp35/50 mucin-specific mAbs.

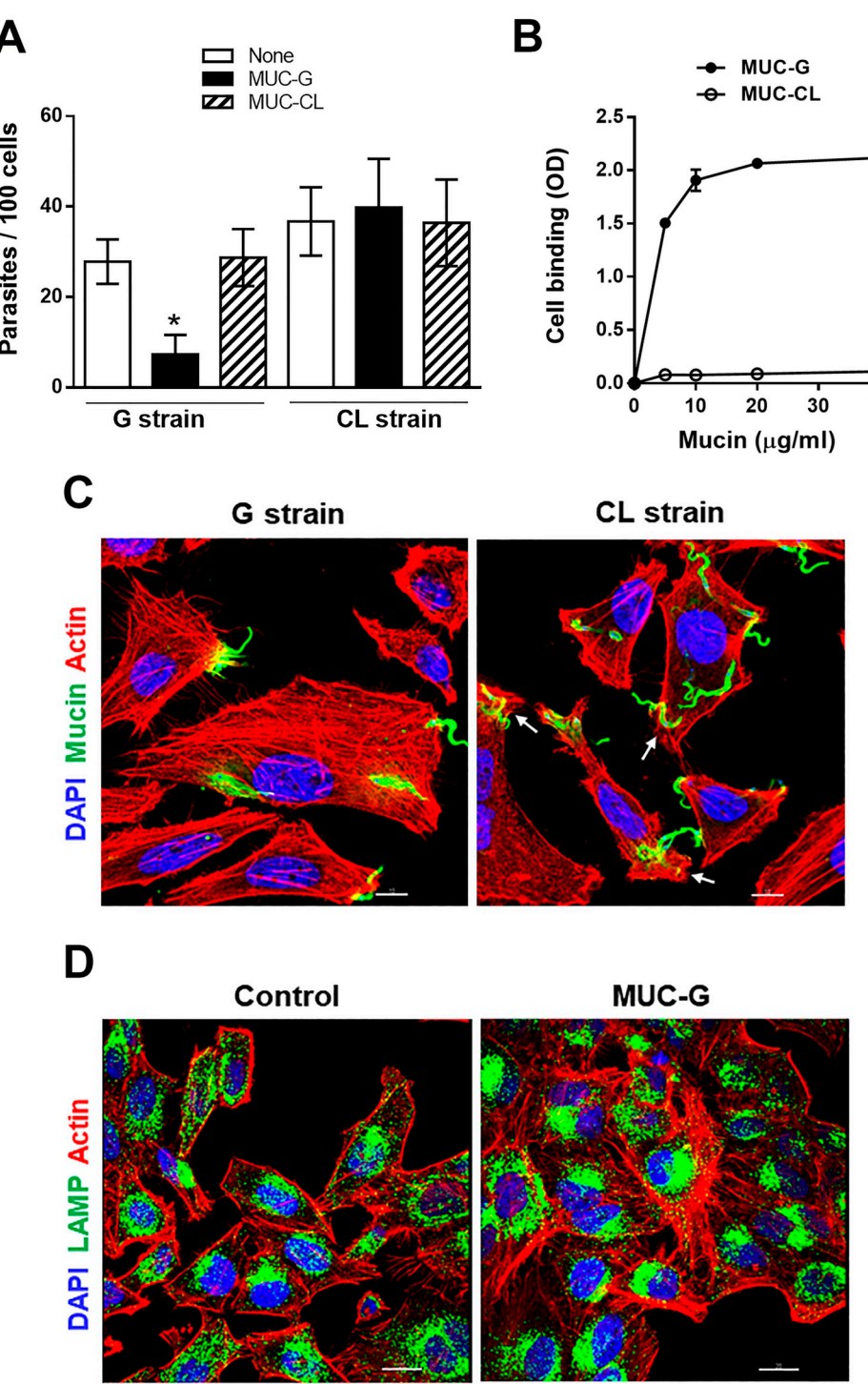

**Fig 2. Inhibition of *T. cruzi* G strain MT internalization and host cell lysosome spreading by purified G-MUC. (A)** HeLa cells were incubated for 1 h with MT of G or CL strain, in absence or in the presence of mucins purified from G strain (G-MUC) or CL strain (CL-MUC), and the number of intracellular parasites was quantified. Values are the means ± four independent assays. Note the significant decrease in G strain MT invasion in the presence of G-MUC (*P<0.001). **(B)** HeLa cells, grown in ELISA microtiter plates, were fixed and incubated for 1 h with MUC-G or MUC-CL at the indicated concentrations. Binding of mucin was revealed by mAb 2B10. The assay was performed in triplicates. **(C)** HeLa cells were incubated for 30 min with G strain or CL strain MT and processed for immunofluorescence analysis for detection of actin (red), nucleus (blue) and mucin (green). Confocal microscopy visualization, under 63x objective, showed thick F-actin filaments in cells incubated with G strain and overall

disruption of F-actin in cells incubated with CL strain. Arrows indicate the site of MT entry with disrupted cortical actin. Scale bar = 10 μm. (**D**) HeLa cells were incubated for 30 min in absence or in the presence of MUC-G and processed for detection of actin (red), nucleus (blue) and lysosome (green). Thick actin bundles were observed in cells incubated with MUC-G. Scale bar = 20 μm.

determine the cell binding capacity of MUC-G and MUC-CL, microtiter plates coated with HeLa cells were incubated for 1 h with MUC-G or MUC-CL, and the binding was revealed using mAb 2B10. Confirming the previous finding [8], MUC-G bound to HeLa cells in a dose-dependent and saturable manner, whereas MUC-CL binding was negligible (Fig 2B). Because it was previously found that treatment of host cells with F-actin disrupting drugs, such as cytochalasin D and latrunculin B, inhibited invasion by G strain, but not by CL strain [10], we examined the architecture of the target cell actin cytoskeleton during MT invasion. HeLa cells were incubated for 30 min with MT and processed for indirect immunofluorescence and confocal microscopy visualization of F-actin and parasites. Well organized thick actin filaments were detected in cells incubated with G strain MT, whereas disrupted F-actin was observed upon incubation with CL strain MT (Fig 2C). Next, the effect of MUC-G on the actin organization was determined. HeLa cells were incubated for 30 min in absence or in the presence of 40 μg/ml MUC-G. Thick actin filament bundles were observed in cells incubated with MUC-G, and we also noted that lysosomes were more densely localized in the perinuclear region, as compared to control cells (Fig 2D). We also checked the effect of MUC-G on lysosome mobilization, HeLa cells were incubated for 30 min in R10 or in PBS$^{++}$, in absence or in the presence of MUC-G or MUC-CL as control. Lysosome spreading induced by PBS$^{++}$ was counteracted by MUC-G, but not MUC-CL (Fig 3). Taken together, these results indicate that the mucin-mediated invasion of G strain MT is actin-dependent and independent of the host cell lysosome mobilization. This is compatible with the finding that the number of LAMP-positive G strain MT is very low, as opposed to the high number of CL strain MT within LAMP-positive parasitophorous vacuole [42].

## Depletion of host cell annexin A2 impairs G strain MT invasion

The gp35/50-mediated invasion of G strain MT is closely associated with target cell actin cytoskeleton, and is inhibited upon treatment with F-actin-disrupting drugs, as reported [10]. We decided to investigate whether annexin A2 was involved in G strain MT invasion, because of its importance as actin nucleator in the vicinity of cellular membranes [5,6]. To that end, we employed the lentiviral transduction methodology to generate Annexin A2-deficient HeLa cells, using two different target sequences. Knockdown (kd) of annexin A2 was detected by western blot analysis in two cell lines (Fig 4A). The susceptibility of these cell lines to MT invasion was checked, by incubating HeLa cells with parasites for 1 h and counting the number of internalized parasites. Both cell lines, annexin A2-kd1 and annexin A2-kd2, were significantly more resistant to G strain MT invasion than wild type (WT) cells (Fig 4B), but remained susceptible to CL strain MT (S2 Fig). By confocal immunofluorescence analysis of uninfected cells, we noted differences between the WT and annexin A2-depleted cells. As compared to WT cells, annexin A2-kd1 cells were much smaller and the clustered lysosomes occupied most part of the cytoplasm, whereas annexin A2-kd2 cells were morphologically similar to WT cells but exhibited more scattered lysosomes (Fig 4C).

## *T. cruzi* G strain mucins interact with target cell annexin A2 and induce focal adhesion kinase (FAK) activation

We examined whether G strain mucins bound to the host cell annexin A2, by co-immunoprecipitation assay. Protein A/G magnetic beads crosslinked with anti-gp35/50 mAb 10D8 were

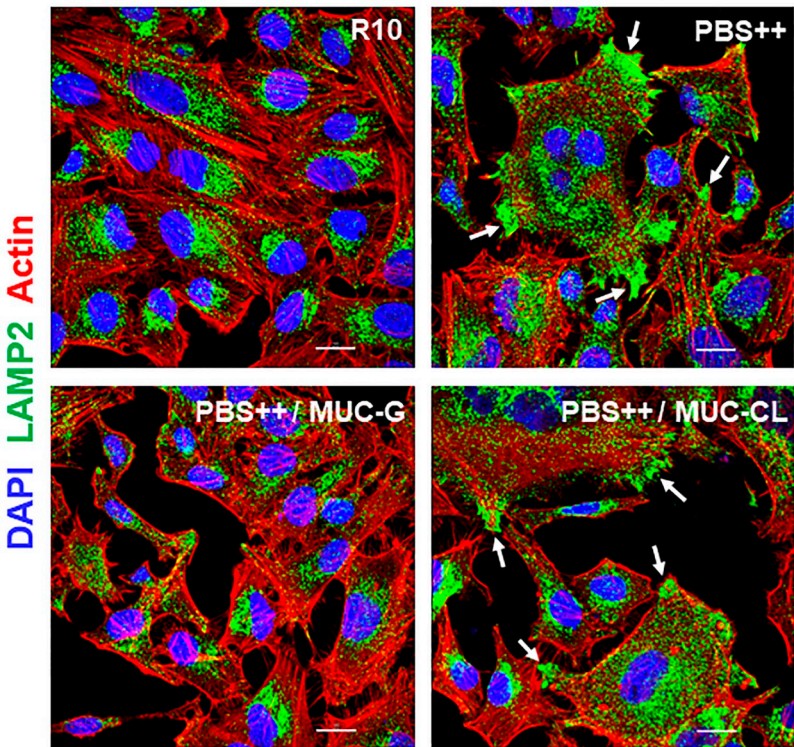

**Fig 3. Inhibition of PBS⁺⁺-induced lysosome spreading by MUC-G.** HeLa cells were incubated for 30 min in RPMI containing 10% serum (R10) or in PBS⁺⁺, in absence or in the presence of 40 µg/ml MUC-G or MUC-CL, and then processed for immunofluorescence analysis for detection of lysosome (green), actin (red) and nucleus (blue) by confocal microscopy. Scale bar = 20 µm. Note the PBS⁺⁺-induced lysosome spreading, with accumulation of lysosomes at the cell edges (white arrows) and inhibition by MUC-G.

incubated for 1 h with G strain MT detergent-soluble extract, washed and then incubated for 1 h with HeLa cell extract. Following washes and elution, the eluted samples were analyzed by western blotting using antibodies to anti-annexin A2 or mAb 10D8. Both annexin A2 and gp35/50 mucins were detected in the co-immunoprecipitated sample (Fig 5A), indicating that mucins do bind to annexin A2. As a previous study showed that protein tyrosine kinase (PTK) inhibition of HeLa cells affected invasion of G strain MT [39], we examined whether purified mucins or MT were capable of inducing PTK activation. HeLa cells were incubated for 30 min in absence or in the presence of MUC-G or with MT, and HeLa cell extracts were analyzed by western blotting, using antibody directed to phosphorylated tyrosine proteins. Tyrosine phosphorylation levels of a 130 kDa protein were increased in cells that interacted with MUC-G or MT, which also induced tyrosine phosphorylation of proteins larger than 180 kDa (S3 Fig). We tested the possibility that FAK might be activated by MUC-G and MT, based on the fact that FAK is a cytoplasmic PTK with a molecular mass of 125 kDa, which regulates the F-actin dynamics [43,44]. HeLa cells were incubated for 30 min in absence or in the presence of 40 µg/ml MUC-G or with MT, followed by western blot analysis, using antibody to phospho-FAK (Tyr397). As shown in Fig 5B, the phosphorylation levels of FAK increased upon HeLa cell interaction with MUC-G or MT. We examined the effect of specific FAK inhibitor on G strain MT internalization. HeLa cells were treated for 30 min with FAK inhibitor PF573228, at varying concentrations. After removal of the drug, the cells were checked for susceptibility to MT invasion, along with untreated control cells. Treatment of HeLa cells with FAK inhibitor, at 40 µM, resulted in significant inhibition of MT internalization (Fig 5C).

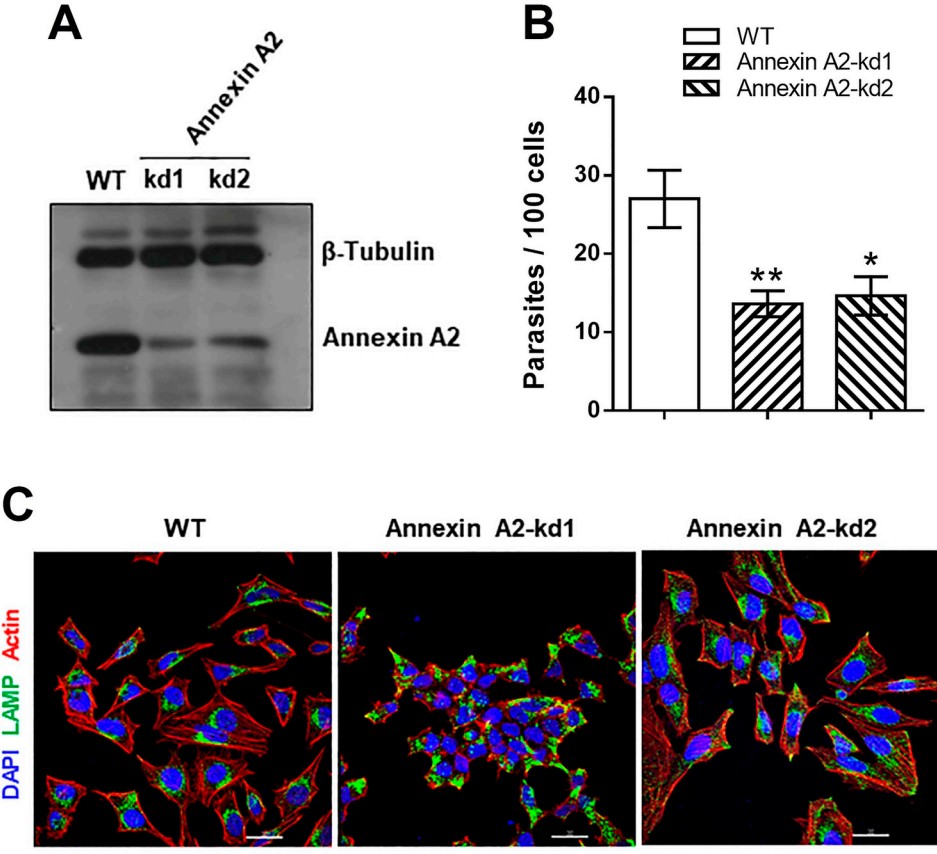

**Fig 4. Increased resistance of annexin A2-depleted cells to G strain MT invasion.** (**A**) HeLa cells were submitted to lentiviral transduction for annexin A2 knockdown (kd) and analyzed by western blotting. Note the depletion of annexin A2 in two independent cell lines. (**B**) HeLa cells depleted in annexin A2 and WT cells were incubated for 1 h with G strain MT. The amounts of intracellular parasites are shown as means ± SD of three independent assays performed in duplicate. MT invasion was significantly diminished in cells deficient in annexin A2 (*P<0.01, **P<0.005). (**C**) Non infected annexin A2-deficient and WT cells were analyzed by immunofluorescence, to visualize actin (red), lysosome (green) and nucleus (blue). Scale bar = 30 μm. Note the altered morphology of annexin-kd1 cells, as compared to WT cells, and the distinct lysosome distribution in annexin-kd2 cells.

## Invading G strain MT colocalizes with clathrin and inhibition of clathrin-coated pit formation decreases parasite internalization

As clathrin-mediated endocytosis is associated with actin cytoskeleton that participates in membrane dynamics [17], we examined whether there was any association of invading G strain MT with clathrin. HeLa cells were incubated with G strain MT for 30 min in PBS$^{++}$, and then processed for immunofluorescence microscopy, using anti-clathrin antibody and mAb 10D8 for detection of parasites. Clathrin-positive parasites were visualized (Fig 6A and 6B). In addition to the colocalization of clathrin with MT, the magnified image of a single cell revealed the colocalization of clathrin with shed mucins at the cell surface (Fig 6B). These are probably gp35/50 mucins that are spontaneously released by G strain MT into medium and bind to annexin A2 at the cell membrane. Next, we checked the effect of treatment of HeLa cells with sucrose on G strain invasion because the treatment of macrophages and epithelial cells with sucrose at 0.45 M was reported to inhibit the formation of clathrin-coated vesicles and to reduce the internalization of Y strain TCT and EA [16]. HeLa cells were treated for 30 min

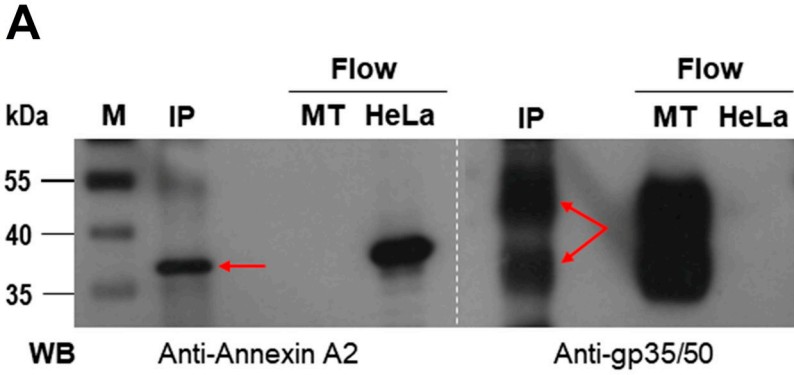

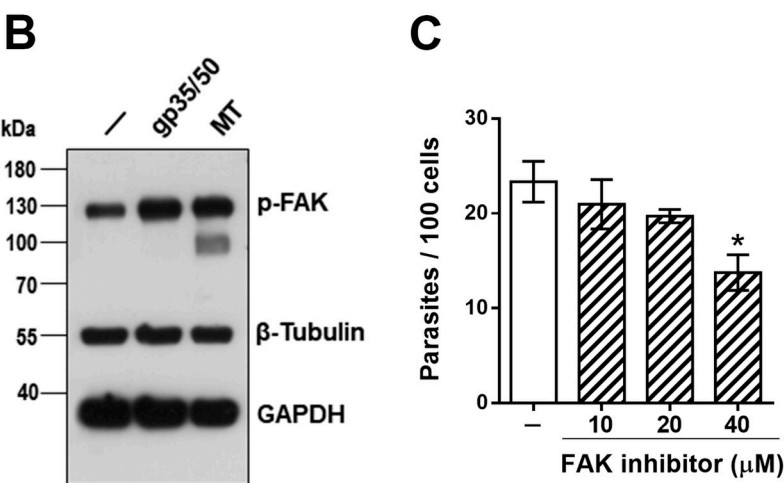

**Fig 5. Binding of G strain mucins to annexin A2 and activation of host cell FAK.** (**A**) Protein A/G magnetic beads, crosslinked to mAb 10D8 directed to gp35/50 mucins, were incubated for 1 h with MT lysate and afterwards with HeLa cell extract for 1 h. The eluate corresponding to immunoprecipitate (IP) was analyzed by western blot (WB), along with the flowthrough samples corresponding to HeLa cell extract or MT. The blot was revealed with anti-annexin A2 antibody or with mAb 10D8. Note that both annexin A2 and gp35/50 mucins were detected in IP from beads crosslinked to mAb 10D8 (red arrows). (**B**) HeLa cells were incubated for 30 min in absence or in the presence of MT or purified gp35/50 mucins at 40 µg/ml. After washings, the cell extracts were analyzed by western blotting, using antibody directed to phosphorylated FAK. Note the increased phosphorylation levels of FAK in cells that interacted with MT or with gp35/50 mucins. (**C**) HeLa cells, untreated or pretreated with FAK inhibitor at indicated concentrations, were incubated for 1 h with MT and the number of internalized parasites was counted. Values are means ± SD of three independent assays performed in duplicate. MT invasion was significantly reduced in cells pretreated with 40 µM ($^*P < 0.005$).

with 0.45 M sucrose in serum-free medium, washed and incubated for 1 h with G strain MT in PBS$^{++}$ or with EA in R10. Cells pretreated with sucrose were significantly more resistant to MT as well as to EA invasion (Fig 7A and 7B). To determine the pattern of MT-clathrin association upon sucrose treatment, immunofluorescence analysis was performed with untreated and sucrose-treated HeLa cells incubated for 30 min with MT. Fewer clathrin-positive parasites were detected in sucrose-treated cells, as compared to untreated controls (Fig 7C). Images of selected fields, showing comparable number of parasites, revealed in untreated cells more spots of clathrin-positive gp35/50 mucins, either in the vicinity of parasites or attached to cell membrane (Fig 8). In addition to treatment with sucrose, HeLa cells were subjected to other procedures, such as treatment with chlorpromazine, intracellular K$^+$ depletion or cytoplasmic

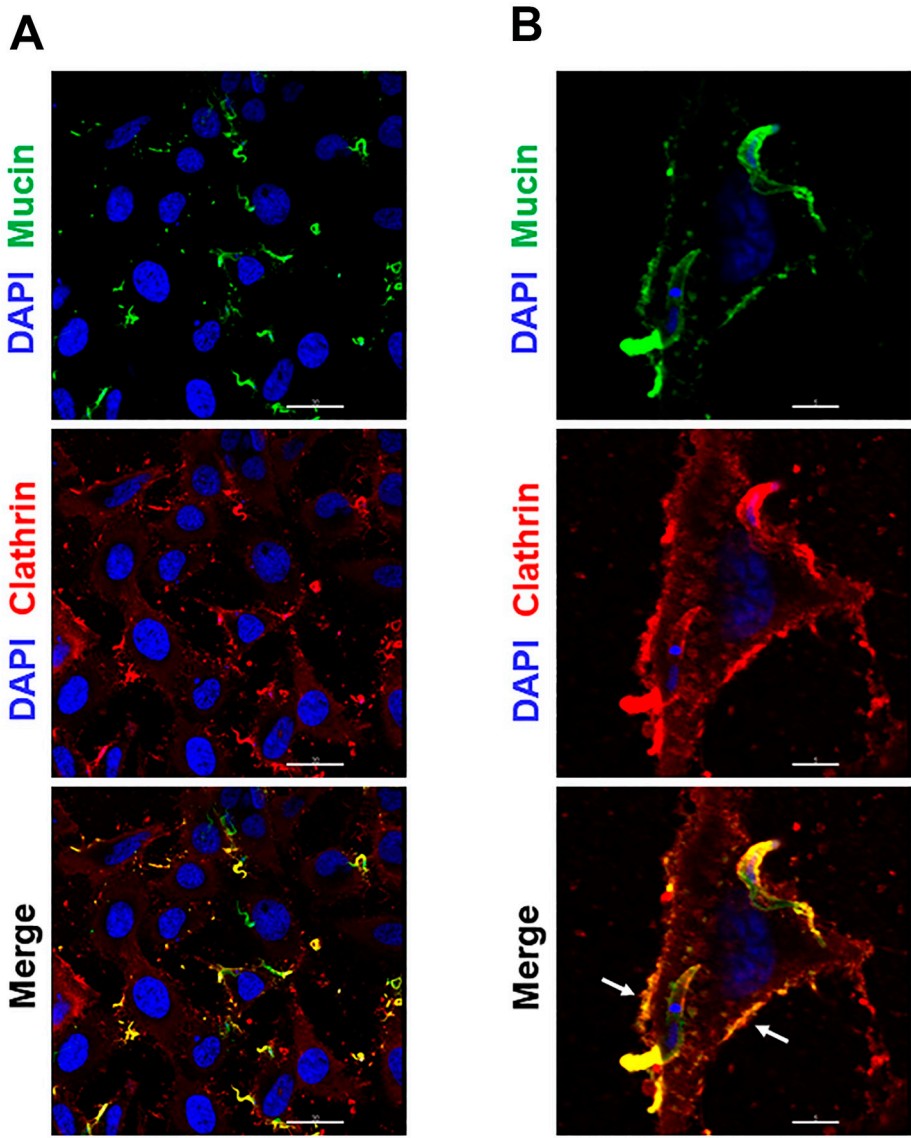

**Fig 6. Colocalization of invading G strain MT and released gp35/50 mucins with host cell clathrin.** (**A**) HeLa cells were incubated for 30 min with parasites and processed for immunofluorescence microscopy, using anti-clathrin antibody and mAb 10D8. The images show clathrin (red), parasite mucins (green) and nucleus (blue). Scale bar = 30 μm. Note the colocalization of MT with clathrin. (**B**) A single cell with invading parasites is depicted to show the colocalization of clathrin with MT and also with shed gp35/50 mucins at the cell membrane (white arrows). Scale bar = 5 μm.

acidification, reported to affect clathrin-coated pits and endocytosis [16,37,38]. HeLa cells, subjected to the referred procedures, as detailed in Methods, were incubated for 1 h with MT, and the number of internalized parasites was counted. These cells were significantly more resistant to MT invasion, as compared to control cells (S4 Fig).

## G strain MT invasion does not require gp82

To determine the participation of gp82 on G strain MT invasion, the effect of mAb 3F6 directed to gp82 was examined. Parasites were pretreated for 30 min with ascitic fluid

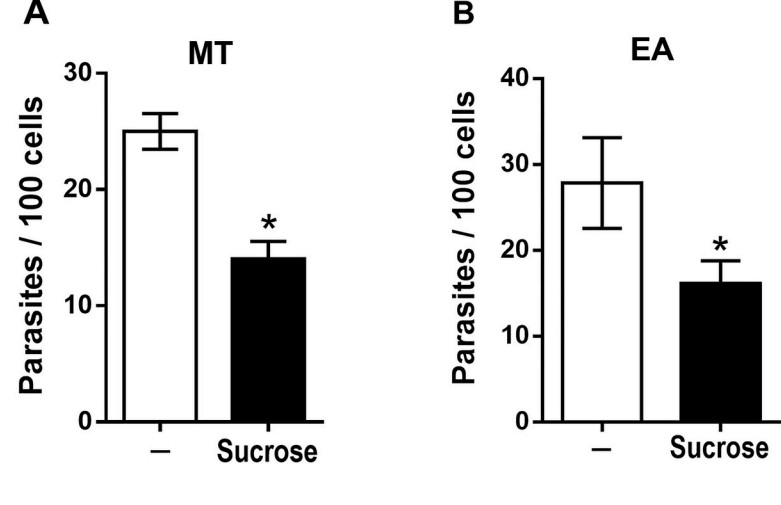

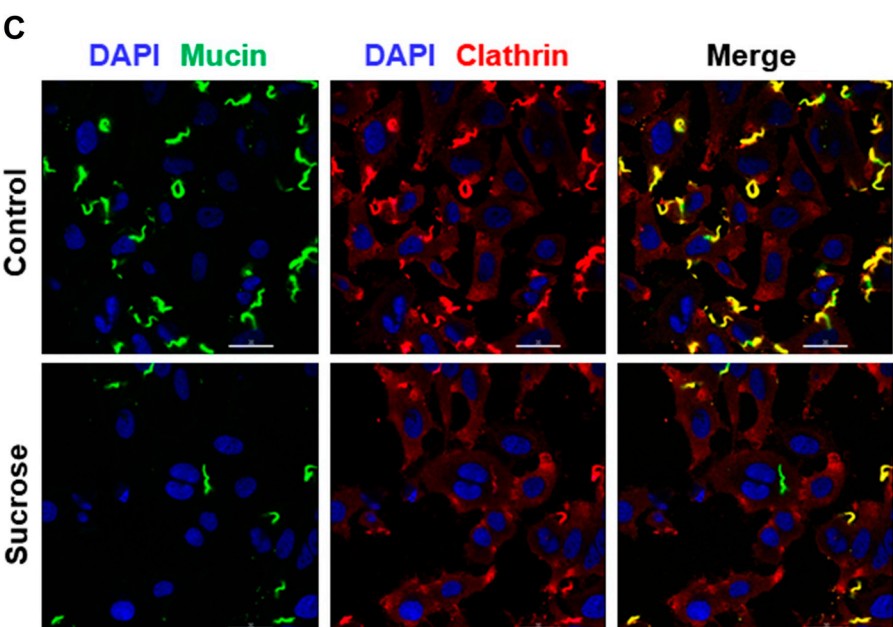

**Fig 7. Inhibition of G strain MT invasion by pretreatment of host cells with sucrose.** (**A**) HeLa cells were treated for 30 min with 0.45 M sucrose in serum-free medium, washed and incubated with: (**A**) G strain MT in PBS$^{++}$ or (**B**) G strain EA in R10. After 1 h, the cells were processed for intracellular parasite quantification. Values are the means ± five independent assays. HeLa cells pretreated with sucrose were significantly more resistant to invasion by MT (*P<0.0001) or EA (*P<0.005). (**C**) Untreated and sucrose-treated HeLa cells were incubated for 30 min with MT and processed for immunofluorescence microscopy, using anti-clathrin antibody and mAb 10D8. The images show clathrin (red), parasite mucins (green) and nucleus (blue). Scale bar = 30 μm.

containing mAb 3F6 or mAb 5E7 that does not recognize live MT, at 1:100 dilution, and then were incubated for 1 h with HeLa cells. Quantification of intracellular parasites showed a significantly augmented invasion capacity of MT pretreated with mAb 3F6 (Fig 9A). In addition, an experiment with purified mAb 3F6 was performed. Parasites were pretreated for 30 min with mAb 3F6, at two different concentrations, and then used for invasion assay. A significant increased MT invasion resulted upon incubation of parasites with mAb 3F6 at 200 μg/ml (Fig 9B). The effect of recombinant gp82 protein (r-gp82), prepared as detailed [21], was also

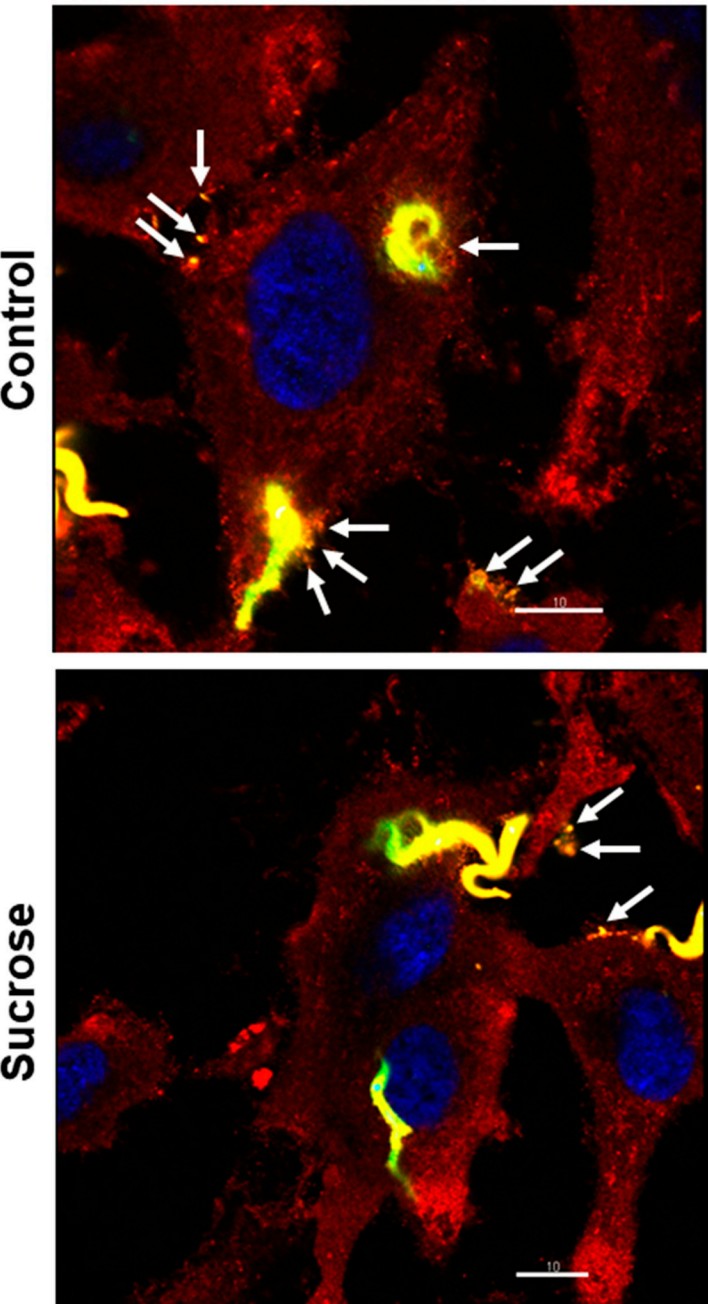

**Fig 8. Reduced gp35/50 mucin co-localization with clathrin in sucrose-treated cells.** Untreated and sucrose-treated HeLa cells were incubated for 30 min with MT and processed for immunofluorescence microscopy, using anti-clathrin antibody and mAb 10D8. Note the clathrin-positive gp35/50 mucins (arrows) in the parasite vicinity or attached to cell membrane. Scale bar = 10 μm.

checked. HeLa cells were incubated for 1 h in absence or in the presence of 40 μg/ml r-gp82 or GST, used as control for the recombinant protein fused to GST, and the number of internal-ized parasites was quantified. MT invasion was significantly inhibited by r-gp82 but not by GST (Fig 9C). Presumably the inhibitory effect of r-gp82 is due to its ability to induce F-actin

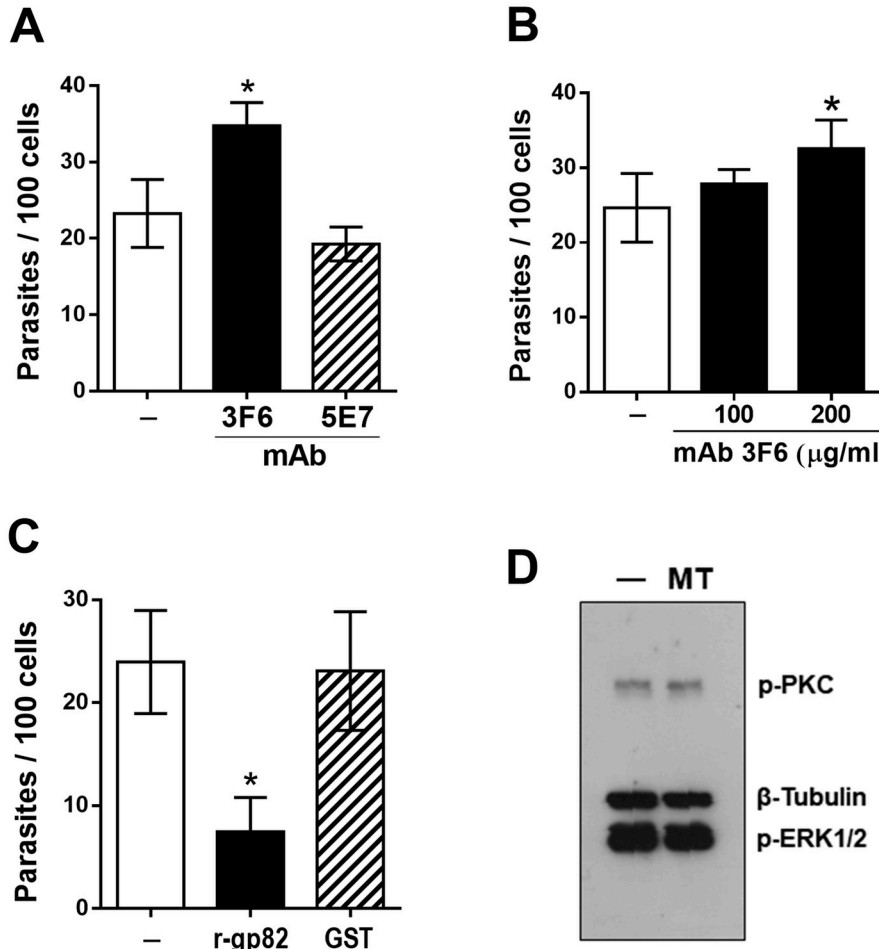

**Fig 9. Lack of requirement of gp82 in G strain MT invasion.** HeLa cells were incubated for 1 h with: (**A**) MT pretreated for 30 min with non purified anti-gp82 mAb 3F6 or mAb 5E7, or (**B**) purified mAb 3F6 at indicated concentrations, (**C**) MT in absence or in the presence of recombinant gp82 protein or GST, and processed for intracellular parasite quantification. MT invasion was significantly increased by treatment with non purified mAb 3F6 or by purified antibody at 200 μg/ml (*P<0.05), and decreased in the presence of r-gp82 (*P<0.005). (**D**) HeLa cells were incubated for 30 min in absence or in the presence of MT. After washings, the cell extracts were analyzed by western blotting, using antibody directed to phosphorylated PKC or ERK1/2. Note that incubation with MT did not alter the phosphorylation levels of PKC or ERK1/2.

disruption and lysosome spreading. These results indicate that G strain MT invasion does not require gp82, which plays a major role in CL strain internalization [23]. As gp82-mediated CL strain invasion is associated with activation of the target cell protein kinase C (PKC) and extra-cellular signal-regulated protein kinases (ERK1/2) [25], an assay was peformed in which HeLa cells were incubated for 30 min with G strain MT and the soluble cell extract was analyzed by western blotting to check the phosphorylation levels of PKC and ERK1/2, as compared to cells incubated in absence of parasites. Activation of PKC or ERK1/2 was not detected, as shown by unaltered phosphorylation levels in cells incubated with MT (Fig 9D).

## Discussion

Gp35/50 mucins, which are highly glycosylated and resistant to proteolytic degradation [34,36], play an important role in oral *T. cruzi* infection, which is currently a frequent mode of

transmission. Studies in mice have shown that metacyclic forms invade the gastric mucosal epithelia as a portal of entry for systemic infection [45,46]. The resistance of MT of different *T. cruzi* strains against peptic digestion is presumably conferred by gp35/50 mucins. When recovered from the mouse stomach 1 h after oral inoculation, the infectivity of MT of different strains towards HeLa cells was fully preserved, similarly to the MT ability to invade cells upon treatment with pepsin at acidic pH [47]. Differently from MT, bloodform trypomastigotes cannot efficiently initiate mucosal infection after an oral challenge [48] and this may be associated with the partial susceptibility of mammalian stage trypomastigotes to some proteases [33].

The present study provides robust evidences that gp35/50 mucins mediate the host cell invasion of G strain MT and disclosed the part played by annexin A2 as the receptor for gp35/50. We detected the interaction of annexin A2 with the native gp35/50 mucins and found that depletion of annexin A2 reduces the host cell susceptibility to G strain MT internalization. Consistent with the involvement of annexin A2, which is known to be recruited to actin assembly sites at cellular membranes [4–6], thick F-actin bundles were observed in cells upon interaction with G strain MT or with purified G strain mucins. That G strain MT exploits the clathrin-mediated endocytic pathway was indicated by detection of clathrin colocalized with invading parasites, and by decreased internalization upon host cell treatment that inhibits clathrin-coated vesicle formation. Such a mechanism resembles that used by *Listeria monocytogenes*, which relies on the interaction of the surface protein InlB with its receptor Met, to invade cells through the clathrin-dependent endocytic pathway [49].

Gp35/50-mediated G strain MT invasion bears similarities to EA internalization, although gp35/50 mucins are not expressed in EA and the two parasite forms interact with different structures on HeLa cell surface, MT invading cells preferentially at the edges and EA binding to surface microvilli [50]. Like G strain MT internalization, EA entry into host cells is an actin-dependent and clathrin-mediated endocytic process, which is impaired in annexin A2-depleted cells [14–16], and lysosomes are not required. At the early stages of EA invasion, lysosomes are concentrated in the perinuclear area, actin recruitment is induced and actin-rich small cup-like membrane protrusions are seen around invading EAs [13,14]. Such structures, comparable to actin-rich pedestals induced by enteropathogenic *Escherichia coli*, which recruits annexin A2 to the bacterial attachment site [51], are not formed during MT invasion. Of note is that in a mixed infection, when G strain MTs were added to HeLa cells shortly before enteroinvasive *E. coli* (EIEC), the bacterial uptake was significantly increased, similarly to what is seen when purified gp35/50 mucins were added [10]. By contrast, EIEC uptake by HeLa cells was inhibited in the presence of CL strain MT or the recombinant gp82 protein [10].

Our data indicated that gp82, which plays a major role in CL strain MT invasion is not required for G strain MT internalization, although the gp82 molecule expressed in the two strains share 100% identity as regards the domain containing the cell binding site [23]. Anti-gp82 mAb 3F6, which inhibits CL strain invasion [30,52], had an opposite effect on G strain internalization. It is conceivable that, when gp82-mediated MT interaction is impaired by mAb 3F6, the host cell binding of gp35/50 is facilitated, because of appropriate F-actin arrangement and lysosome distribution. In accord is the finding that the recombinant gp82 exerts an opposite effect. It promotes F-actin disruption and lysosome scattering, unfavorable to gp35/50-dependent MT internalization. We presume that in G strain MT-host cell interaction binding of 35/50 mucins prevails over that of gp82, because of their abundance and because annexin A2 is distributed throughout cell membrane, whereas the gp82 receptor LAMP2, associated with the lysosome membrane, is expressed at low levels on the cell surface. Gp82-mediated invasion of CL strain induced the activation of PKC and ERK1/2 in host cells [25]. Compatible with the lack of gp82 participation in G strain MT invasion, neither PKC nor ERK1/2 was activated upon parasite-host cell interaction. The signaling pathway triggered

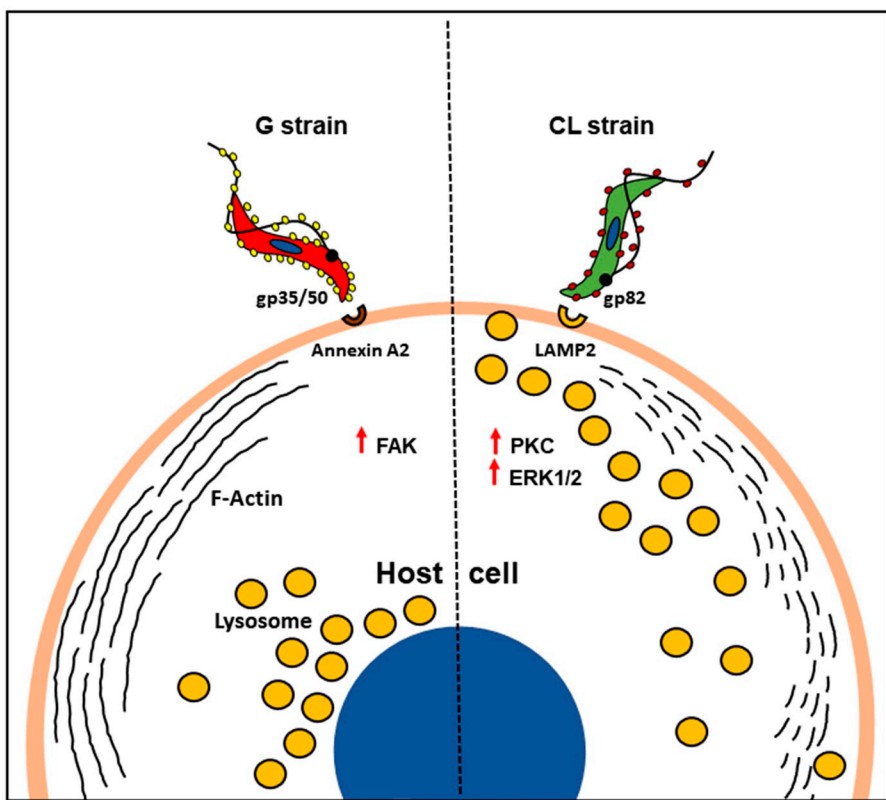

**Fig 10. Schematic representation of possible mechanisms of host cell invasion by *T. cruzi* strains G and CL.** Host cell invasion by G strain MT involves the interaction of gp35/50 mucins with annexin A2, FAK activation and F-actin recruitment, whereas CL strain MT internalization relies on the recognition of gp82 by LAMP2, activation of PKC and ERK1/2, disruption of F-actin and spreading of lysosomes to the cell periphery.

during gp35/50-mediated MT internalization appears to involve PTK, including FAK. For host cell invasion by CL strain MT, PTK is not required [39]. From the available data, plus the new informations from the present study, we can visualize a more clear picture of the differences between G and CL strains, as regards the mechanisms of host cell entry (Fig 10). Invasion of G strain MT is mediated predominantly by gp35/50 mucins that interact with host cell annexin A2, induce PTK activation and recruitment of F-actin. On the other hand, CL strain MT internalization involves the recognition of gp82 by its receptor LAMP2, the activation of PKC and ERK1/2, disruption of F-actin and lysosome spreading to the cell periphery.

## Supporting information

**S1 Fig. Effect of incubation medium on release of gp35/50 mucins and cell invasion capacity of G strain MT.** (A) Conditioned medium from parasites, incubated for 30 min in RPMI medium containing 1% serum (R1) or in PBS$^{++}$, was analyzed by western blotting using mAb 10D8. Note the higher mucin levels in R1. (B) HeLa cells were incubated for 1 h with MT in R1 or PBS$^{++}$, and the number of intracellular parasites was quantified. Values are the means ± five independent assays.
(TIF)

**S2 Fig. Unchanged susceptibility of annexin A2-depleted cells to CL strain MT invasion.** HeLa cells depleted in annexin A2 and WT cells were incubated for 1 h with CL strain MT.

The amounts of intracellular parasites are shown as means ± SD of three independent assays performed in duplicate.
(TIF)

**S3 Fig. Activation of host cell PTK by MUC-G and G strain MT.** HeLa cells were incubated for 30 min in absence or in the presence of MT or MUC-G at 40 μg/ml. After washings, the cell extracts were analyzed by western blotting, using antibody directed to phosphorylated tyrosine proteins. Note the increased phosphorylation levels of protein bands in cells that interacted with MT or with MUC-G (black arrows).
(TIF)

**S4 Fig. Effect of diverse treatments of host cell on G strain MT invasion.** HeLa cells were subjected to chlorpromazine treatment, intracellular $K^+$ depletion or cytoplasmic acidification, and then incubated with MT for 1 h, followed by internalized parasite quantification. Values are the means ± three independent assays performed in duplicate. MT invasion was significantly reduced in cells subjected to diverse procedures (*P<0.001, **P<0.0005).
(TIF)

## Author Contributions

**Conceptualization:** Thiago Souza Onofre, Nobuko Yoshida.

**Formal analysis:** Thiago Souza Onofre, Nobuko Yoshida.

**Investigation:** Thiago Souza Onofre, Leonardo Loch, João Paulo Ferreira Rodrigues, Nobuko Yoshida.

**Methodology:** Thiago Souza Onofre, Silene Macedo.

**Supervision:** Nobuko Yoshida.

**Writing – original draft:** Nobuko Yoshida.

**Writing – review & editing:** Thiago Souza Onofre, Leonardo Loch, João Paulo Ferreira Rodrigues, Silene Macedo, Nobuko Yoshida.

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
