## [Decision Letter · Decision Letter 0]

7 Jun 2022

Dear Prof. Yoshida,

Thank you very much for submitting your manuscript "Mucin molecules of Trypanosoma cruzi metacyclic forms that mediate host cell invasion interact with annexin A2" for consideration at PLOS Neglected Tropical Diseases. As with all papers reviewed by the journal, your manuscript was reviewed by members of the editorial board and by several independent reviewers. In light of the reviews (below this email), we would like to invite the resubmission of a significantly-revised version that takes into account the reviewers' comments.

We cannot make any decision about publication until we have seen the revised manuscript and your response to the reviewers' comments. Your revised manuscript is also likely to be sent to reviewers for further evaluation.

Sincerely,

Igor C. Almeida

Associate Editor

Margaret Phillips

Deputy Editor

Reviewer's Responses to Questions

**Key Review Criteria Required for Acceptance?**

**Methods**

-Are the objectives of the study clearly articulated with a clear testable hypothesis stated?

-Is the study design appropriate to address the stated objectives?

-Is the population clearly described and appropriate for the hypothesis being tested?

-Is the sample size sufficient to ensure adequate power to address the hypothesis being tested?

-Were correct statistical analysis used to support conclusions?

-Are there concerns about ethical or regulatory requirements being met?

Reviewer #1: -Are the objectives of the study clearly articulated with a clear testable hypothesis stated? YES

-Is the study design appropriate to address the stated objectives? MOSTLY YES

-Is the population clearly described and appropriate for the hypothesis being tested? N/A

-Is the sample size sufficient to ensure adequate power to address the hypothesis being tested? YES

-Were correct statistical analysis used to support conclusions? YES

-Are there concerns about ethical or regulatory requirements being met? NO

Reviewer #2: The aims of the study are articulated with a testable hypothesis. The study design is appropriate to address the stated objectives and it is strongly supported by previous studies carried out by the same authors. The statistical analysis seems correct and supports conclusions. The authors analyzed data by t-test. Most assays incorporate two or three groups (control vs treatments) and comparisons were performed in pairs (control vs treatment 1 or treatment 2). The study was approved by a Committee on Ethics of Animals despite not incorporating in vivo assays. An approved biosecurity certificate should be incorporated.

Reviewer #3: -Are the objectives of the study clearly articulated with a clear testable hypothesis stated?

Yes

-Is the study design appropriate to address the stated objectives?

Yes, with some concerns. See Results section below.

-Is the population clearly described and appropriate for the hypothesis being tested?

-Is the sample size sufficient to ensure adequate power to address the hypothesis being tested?

Yes

-Were correct statistical analysis used to support conclusions?

Yes 

-Are there concerns about ethical or regulatory requirements being met?

yes

**Results**

-Does the analysis presented match the analysis plan?

-Are the results clearly and completely presented?

-Are the figures (Tables, Images) of sufficient quality for clarity?

Reviewer #1: -Does the analysis presented match the analysis plan? NO EXPLICIT ANALYSIS PLAN; METHODS AND RESULTS WELL ALIGNED THOUGH

-Are the results clearly and completely presented? YES

-Are the figures (Tables, Images) of sufficient quality for clarity? YES

Reviewer #2: The analysis presented matches the analysis plan. The results are completely presented, however, there are some points which should be clarified to improve the understanding of the article by readers (explained below). The figures do not have sufficient quality. The resolution level of images (microscopy) should be improved. Additionally, there are four supplementary figures. Some of these (S2 and S4) are important to support the hypothesis, therefore I suggest incorporating them into the article’s figures.

Reviewer #3: From a cell biology point of view, the experiments are very preliminary and little explored even for a first approach. For instance, the authors detected high MW phosphorylated proteins but no efforts were made to identify them (Fig 4). Similarly, the use of 0.45M sucrose to analyze the association of clathrin with the events under study is a coarse conception and requires further analysis with more appropriate inhibitors (http://dx.doi.org/10.4161/cl.23967).

**Conclusions**

-Are the conclusions supported by the data presented?

-Are the limitations of analysis clearly described?

-Do the authors discuss how these data can be helpful to advance our understanding of the topic under study?

-Is public health relevance addressed?

Reviewer #1: -Are the conclusions supported by the data presented? MOSTLY YES

-Are the limitations of analysis clearly described? NOT EXPLICITLY

-Do the authors discuss how these data can be helpful to advance our understanding of the topic under study? PARTIALLY YES

-Is public health relevance addressed? NOT EXPLICITLY

Reviewer #2: The conclusions are supported by the data presented. Some supplementary material could be incorporated as Figures. The limitations of analysis are described, but the most of M&M is supported by previous published data. I suggest explicitly mentioning how these data can be helpful to advance our understanding of the topic under study and its relevance in public health in the discussion section.

Reviewer #3: The use of hypertonic media to test for inhibition of clathrin-coated vesicles is only a first approach. More appropriate inhibitors should be tested to confirm that the internalization of the G strain is really due to this event and not to the strong stress that this technique exerts on the host cell.

**Editorial and Data Presentation Modifications?**

Reviewer #1: See general comments

Reviewer #2: The resolution of figures, mainly microscopy images, should be improved.

Reviewer #3: Fig 1B: Please replace G-strain with MUC-G and CL strain with MUC-CL.

**Summary and General Comments**

Reviewer #1: The paper provides new insights into the molecular mechanism of in vitro host cell invasion by T. cruzi. The data are an important, though incremental, advance on a long series of studies from this lab focussing on the differences between well charaterised T. cruzi strains and their surface proteins. Experiment design was robust, the data presentation is very clear and the ms is well written. I have a few concerns/queries about interpretation of some of the experiments and there are areas where some more explanation would help the generalist reader follow the paper.

1. The introduction would benefit from some further explanation of what genes encode gp35/50 and gp82 and how that relates to the genomes/transcriptomes/proteomes of G and CLBR strains, or an indication of where there are gaps in this kind of knowledge. Presumably these are multicopy genes, so is it known how many copies are in each strain? For example, it is not clear whether the G strain has gp82 genes or not, or if it has them then are they expressed (in trypos?) or are they very divergent sequences vs CLBR or have different glycosylation function. Likewise for gp35/50 genes in CLBR. 

2. Purification of mucins used 5 × 10^10 “parasites”, but which forms were used is not explicit? Presumably they were epimastigotes though, in which case it should be clarified what is known about gp82 and gp35/50 expression in epimastigotes. This would help with the interpretation of figure 2.

3. Perhaps a naïve biochemistry question, but how can we be sure that the F3 fraction contains purely mucins? The authors state “We followed the protocol used previously to purify mucins from different T. cruzi strains [32=Maeda et al 2011].” However, the Maeda paper references the protocol further back to a review article (Acosta-Serrano et al 2001) that does provide any more details about the protocol or its validation.

4. Immunofluorescence images (Figures 1,2,3 etc) are all reported qualitatively. The phenotypes look convincing enough so quantitative image analysis would probably be overkill, but I think the methods and or figure legends should indicate how many fields of view were checked for consistency with these representative images.

5. Why/how did they decide to focus specifically on Annexin A2? I think a line of explanation is needed at the start of the section at Line 281.

6. Figure 3C and related text – should clarify if these are uninfected or infected cells.

7. Line 325, epitelial -> epithelial

8. Figure 6, the specificity of the sucrose pre-treatment to clathrin-mediated endocytosis seems questionable, so I think the data on parasite internalisation in this figure should be supported by the clathrin and mucin immunofluorescence assays as per figure 5. This should reveal/confirm the extent to which the sucrose has affected calthrin-parasite/mucin co-localisation.

9. Line 332 “G strain MT invasion does not require gp82”. I’m not sure that the experiment allows such a definitive statement to be made, I think they would need to delete the gene to prove it is not required. It would be useful to know whether antibody concentration is a factor (is the augmentation dose-dependent?), the amount used is not stated. Also, according to the reference #30 (Teixeira et al 1986), the Mab 3F6 binds proteins of MW 82kDa and 75kDa, so how can specific inferences be made about gp82? Discussion relating to these data could be re-phrased for the same reasons. 

10. Line 383, date -> data

11. I think the discussion would benefit from some comments on the importance of the results in the context of the broader diversity of T. cruzi, the potential difference between metacyclics and blood/tissue culture trypos and the cell tropism in vivo (how representative are HeLa cells?)

Reviewer #2: This study explores the mechanisms involved in metacyclic trypomastigotes from G strain (TcI) entry into the host cell mediated by gp35/50, a mucin-like glycoprotein, expressed on Trypanosoma cruzi. As specific aims, the article proposes to decipher the host cell receptor involved in this interaction and an approach to the intracellular mechanisms activated. The study is novel and significant to contribute to the knowledge about the mechanisms of infectivity involved in Trypanosoma cruzi, the causal agent of Chagas disease, a NTDs with a complex life cycle and biology. The results obtained herein are strengths and strongly supported by a methodology standardized and published by the authors in previous studies. However, some points could be clarified to improve the understanding of the study.

Specific comments:

- In the M & M section, authors describe the purification of T. cruzi mucin-like molecules. In line 142, after “A total of 5 x 10^10 parasites”, I suggest to add the strains of parasites used (G and CL strains).

- In this point (line 157-158), authors indicate that “fractions were analyzed by silver staining of SDS-PAGE gels, as well as by immunoblotting using the available monoclonal antibodies”. Were these analyses used to identify different mucins present in these fractions (gp35/50, gp90 and gp82). If it is possible, I suggest, to add this characterization to the results and the proportion of specific mucins in MUC-G and MUC-CL. In case this characterization has been published previously, I suggest incorporating a reference.

- In some sections of results, authors point out a purified gp35/50 (line 307). It is not completely clear if this purified gp35/50 is a synonym of MUC-G. Please clarify.

- Figure 8. The schematic representation of possible mechanisms of host cell invasion by T. cruzi strains G and CL, is a good figure that summarizes the main results. However, some important information highlighted here is presented in supplementary material. In this sense, I suggest incorporating S2 and S4 into the article’s figures.

- The abstract, summary and results are focused mainly in gp35/50. Therefore, I recommend replacing “mucin molecules” by gp35/50 in the title.

- The resolution of figures should be improved.

- The study was approved by a Committee on Ethics of Animals despite not incorporating in vivo assays. Please, add a biosecurity certificate approved.

Reviewer #3: The report is well written and easily followed, leading to confirmation of previous findings and further supporting knowledge of parasite's invasion strategies.

However, the observation of protein phosphorylation without involvement of the affected proteins together with the poor focus of the hypertonic medium strongly cloud the report.

To fully support the involvement of clathrin-coated vesicles in parasite invasion, several assays with appropriate inhibitors must be performed.

PLOS authors have the option to publish the peer review history of their article (what does this mean?). If published, this will include your full peer review and any attached files.

Reviewer #1: No

Reviewer #2: No

Reviewer #3: No
---

## [Decision Letter · Decision Letter 1]

5 Sep 2022

Dear Prof. Yoshida,

We are pleased to inform you that your manuscript 'Gp35/50 mucin molecules of Trypanosoma cruzi metacyclic forms that mediate host cell invasion interact with annexin A2' has been provisionally accepted for publication in PLOS Neglected Tropical Diseases.

Best regards,

Igor C. Almeida

Academic Editor

Margaret Phillips

Section Editor

Reviewer's Responses to Questions

**Key Review Criteria Required for Acceptance?**

**Methods**

-Are the objectives of the study clearly articulated with a clear testable hypothesis stated?

-Is the study design appropriate to address the stated objectives?

-Is the population clearly described and appropriate for the hypothesis being tested?

-Is the sample size sufficient to ensure adequate power to address the hypothesis being tested?

-Were correct statistical analysis used to support conclusions?

-Are there concerns about ethical or regulatory requirements being met?

Reviewer #1: (No Response)

Reviewer #2: -The objectives of the study are clearly articulated with a clear testable hypothesis

-The study was designed appropriate to address the stated objectives

-There are no concerns about ethical or regulatory requirements being met

Reviewer #3: In the revised manuscript, the rigor of the studies carried out has been improved.

**Results**

-Does the analysis presented match the analysis plan?

-Are the results clearly and completely presented?

-Are the figures (Tables, Images) of sufficient quality for clarity?

Reviewer #1: (No Response)

Reviewer #2: -The analysis presented match the analysis plan

-The results are clearly and completely presented

-The figures (Tables, Images) are of sufficient quality for clarity

Reviewer #3: yes

**Conclusions**

-Are the conclusions supported by the data presented?

-Are the limitations of analysis clearly described?

-Do the authors discuss how these data can be helpful to advance our understanding of the topic under study?

-Is public health relevance addressed?

Reviewer #1: (No Response)

Reviewer #2: -The conclusions are supported by the data presented

-The authors discuss how these data can be helpful to advance our understanding of the topic under study

-Public health is relevance addressed

Reviewer #3: yes

**Editorial and Data Presentation Modifications?**

Reviewer #1: (No Response)

Reviewer #2: The authors considered all previous suggestions and comments. The article was improved accordingly. I recommend to Accept the manuscript.

Reviewer #3: Accept

**Summary and General Comments**

Reviewer #1: The changes have greatly improved the ms and addressed all comments and queries.

Reviewer #2: This manuscript explores the mechanisms involved in metacyclic trypomastigotes from G strain (TcI) entry into the host cell mediated by gp35/50 on Trypanosoma cruzi. The authors propose to describe the host cell receptor involved in the gp35/50 interaction and its asociated intracellular mechanisms. The study is novel and significant to contribute to the knowledge about the infectivity process involved in different strains and genotypes of Trypanosoma cruzi, the causal. The results obtained herein are strengths and strongly supported by a methodology standardized and published by the authors in previous studies. The authors considered all previous suggestions and comments. The manuscript was improved accordingly.

Reviewer #3: (No Response)

PLOS authors have the option to publish the peer review history of their article (what does this mean?). If published, this will include your full peer review and any attached files.

Reviewer #1: No

Reviewer #2: **Yes: **Galia Ramírez-Toloza

Reviewer #3: No

---

## [Editor Report · Acceptance letter]

19 Sep 2022

Dear Prof. Yoshida,

We are delighted to inform you that your manuscript, "Gp35/50 mucin molecules of Trypanosoma cruzi metacyclic forms that mediate host cell invasion interact with annexin A2," has been formally accepted for publication in PLOS Neglected Tropical Diseases.

Best regards,

Shaden Kamhawi

co-Editor-in-Chief

Paul Brindley

co-Editor-in-Chief
